

# Classification of spectral fine structures of Saturn kilometric radiation

Georg Fischer[1], Ulrich Taubenschuss[2], and David Píša[2]

[1]Space Research Institute, Austrian Academy of Sciences, Schmiedlstr. 6, A-8042 Graz, Austria
[2]Institute of Atmospheric Physics of the Czech Academy of Sciences, Prague, Czech Republic

**Correspondence:** Georg Fischer (georg.fischer@oeaw.ac.at)

**Abstract.** The spectral fine structures of Saturn kilometric radiation (SKR) are best investigated with the Wideband Receiver (WBR) of Cassini's Radio and Plasma Wave Science (RPWS) instrument, with which measured radio fluxes can be displayed in time–frequency spectra with resolutions of 125 ms and $\sim 0.1$ kHz. We introduce seven different classes of SKR fine structures ranging from dots (one class for 0-dimensional objects) over lines (four classes of 1-dimensional objects being horizontal, vertical, or with negative or positive slope) to areal features (one class for 2-dimensional objects). Additionally, we define a 7th class containing special structures named according to their appearance in time–frequency spectra. These special features are named rain, striations, worms, and caterpillar, and the latter two have never been described in the literature so far.

Using this newly defined classification scheme we classify features in spectra at low frequencies in the baseband of the 80-kHz WBR and at medium frequencies around 325 kHz. A statistics of the occurrence of various classes and sub-classes shows some notable characteristics: Lines with a positive slope are much more common at medium frequencies than at low frequencies, and vertical lines are almost absent at low frequencies. The particular fine structure of striations (group of narrowbanded lines with predominantly negative slopes) is quite common below 80 kHz, but less common near 325 kHz. At these medium frequencies, the lines rather look like interrupted striations, which we term with the name 'rain'. We also find rare instances of striations with a positive slope, and rare instances of absorption signatures within areal features. The newly introduced sub-classes of worms (lines oscillating in frequency) and caterpillar occur almost exclusively below 80 kHz. Caterpillars have a typical bandwidth of $\sim 10$ kHz, a constant frequency below $\sim 40$ kHz for several hours, and they are mostly observed beyond distances of 15 Saturn radii around local dusk. We discuss the implications of our findings in view of the many theories about spectral fine structures of auroral radio emissions.

## 1 Introduction

Auroral radio emissions at Earth and Saturn exhibit a wide variety of fine structure when displayed in dynamic spectra, which show the intensity of the radio waves as a function of time and frequency. It is widely believed that the cyclotron maser



instability (CMI) is the mechanism responsible for the generation of these emissions (Wu and Lee, 1979). Originally the CMI assumed an electron velocity distribution $f$ in the form of a loss cone, but subsequent research (Pritchett and Strangeway, 1985)

and observations with the Viking (Louarn et al., 1990; Roux et al., 1993) or the FAST satellite (Delory et al., 1998) have led to trapped electrons in a ring-shell or horseshoe distribution. Within those distributions the electrons with a positive gradient with respect to the velocity perpendicular to the ambient magnetic field ($\delta f/\delta v_\perp > 0$) do provide the free energy which is converted to the electromagnetic energy of the radio waves.

After the detection of Auroral Kilometric Radiation (AKR) at Earth (Gurnett, 1974) the first report on AKR fine structure

was given by Gurnett et al. (1979). They showed a complex AKR frequency–time structure with many narrowbanded linear emissions drifting mainly upward, but also downward in frequency with time. These drifting features were interpreted as being due to descending or ascending sources along the magnetic field line, respectively. In the following decades a large number of theories were developed, attempting to explain the origin of these fine structures. One of the first models was the one by Calvert (1982) who suggested a feedback or tuned cavity model in which the source region has a density enhancement and acts

as a waveguide with sharp density boundaries and a length equal to a multiple of the wavelength. This was later criticized by Pritchett et al. (2002) as being a model requiring special conditions which are not supported by observations. Melrose (1986) proposed a feedback model depending on a phase-bunching mechanism, in which the speed of the interaction region needs to be intermediate between those of the particles and the waves. Grabbe (1982) explained the special case of banded AKR fine structure separated by the ion cyclotron frequency as direct evidence for a three wave mechanism, in which the AKR is

produced by non-linear interaction between electromagnetic waves and ion cyclotron waves. The latter two models also need special conditions for the creation of the fine structures. The models by McKean and Winglee (1991) and Yoon and Weatherwax (1998) both come to the conclusion that the narrowbandedness of the fine structure of AKR is a natural consequence of the CMI mechanism in a non-uniform magnetic field. McKean and Winglee (1991) reach this conclusion by using one-dimensional particle in cell (PIC) simulations, but their obtained frequency drift rates are of the order of MHz s$^{-1}$, much higher than the

typically observed drifts of a few kHz s$^{-1}$. Yoon and Weatherwax (1998) used a more realistic model of the electron distribution function inside the AKR source region to find that the CMI growth rate has a narrow bandwidth of $\Delta\omega/\omega \approx 10^{-3}$, with $\omega$ being the angular frequency. Pritchett et al. (2002) also reached the same value for the bandwidth by using two-dimensional PIC simulations. Menietti et al. (1996, 2006) specifically investigated the fine structure named striations and suggested a possible stimulation of such AKR by upward traveling electromagnetic ion cyclotron (EMIC) waves. Striations consist of a bundle of

narrowbanded closely spaced negative frequency–drifting signatures with about the same slope corresponding to an upward group velocity of the EMIC waves in the range between 100 and 1000 km s$^{-1}$. Mutel et al. (2006) argued that the striated AKR is consistent with upward traveling ion solitary structures or ion holes. In the last 20 years some theories on the fine structure moved away from the suprathermal electrons as causing the radio emissions and instead suggested electron holes (Pottelette et al., 2001), tripolar structures (Pottelette and Treumann, 2005), ion holes (Mutel et al., 2006), or paired electrons (Treumann

and Baumjohann, 2020). Striations were also found in Cassini spectra of Jovian broadband kilometric radiation (bKOM) by Kurth et al. (2001), but they only termed them as downward drifting fine structures.



At Saturn, the corresponding radio emission to AKR at Earth is Saturn Kilometric Radiation (SKR), which has been extensively observed by Cassini during its 13-year long orbital tour from 2004 until 2017. For SKR observations with high spectral and temporal resolution the Wideband Receiver (WBR) of the Cassini Radio and Plasma Wave Science (RPWS) instrument was used, and it has provided several tens of thousands of spectra with SKR fine structure. Until today, not much from this rich data set has been published. Kurth et al. (2005) presented the first high-resolution dynamic spectra of SKR showing fine structures strikingly similar to AKR at Earth or auroral radio emissions at Jupiter. They observed upward and downward drifting features with bandwidths down to $\sim 200$ Hz and drift rates of a few kHz per second. Menietti and Kurth (2006) identified the ordered fine structure called 'striations', which have been found in Polar or Cluster AKR data, also in SKR. Saturn is also the second planet where a spacecraft has traversed through the auroral radio emission source region. Lamy et al. (2010, 2018) found that the CMI mechanism should be responsible for the SKR generation in which radio waves are amplified perpendicular to the magnetic field by hot electrons in the energy range of 6–12 keV. The SKR sources are located in the upward current region mapping magnetically to the ultraviolet auroral oval. Similar to Earth, the SKR source region is characterized by a ratio of the electron plasma frequency $f_{pe}$ to the electron cyclotron frequency $f_{ce}$ of $f_{pe}/f_{ce} \leq 0.1$. However, there is no terrestrial–like cavity devoid of cold electrons.

Despite the large number of theories mentioned above, no classification scheme for AKR or SKR fine structures has been developed until today. Only for Jovian S-bursts a detailed phenomenological classification and analysis has been made by Riihimaa (1991). In this paper we use the observations of SKR by the Cassini WBR to develop a simple classification scheme with which $\sim 80\%$ of the SKR WBR spectra can be classified. In Section 2 we will describe the main technical characteristics of the Wideband Receiver (WBR) of the Cassini RPWS instrument. The classification scheme containing seven different classes will be introduced in Section 3, along with a detailed description and multiple dynamic spectra as examples. In Section 4 the occurrence probabilities of the various classes will be determined for thousands of arbitrarily chosen SKR wideband spectra for the frequency ranges below 80 kHz and around 325 kHz. Section 5 will discuss some aspects of various classes with respect to physical characteristics or generation hypotheses (e.g., occurrence of areal features and absorption signatures). A brief conclusion and outlook will finish this paper.

## 2 The Wideband Receiver (WBR) of the RPWS instrument

The fine structure of SKR was measured using the Wideband Receiver (WBR) of the Cassini Radio and Plasma Wave Science (RPWS) instrument (Gurnett et al., 2004). It was built at the University of Iowa, and it was similar in design to wideband receivers used on spacecraft like Voyager, Galileo, Polar, or Cluster (Gurnett et al., 1997). On Cassini, the WBR provided high-resolution waveform measurements in passbands of either 60 Hz to 10.5 kHz ('10-kHz wideband') or 0.8 kHz to 75 kHz ('75-kHz wideband'). This was the WBR baseband usage, but it was also possible for the WBR to process signals from the High Frequency Receiver (HFR). Here the HFR down-converted a high frequency signal of 25 kHz bandwidth into a passband from 50 to 75 kHz, which was then sent to the 75-kHz WBR. The HFR could center its 25-kHz band at any frequency between 125 kHz and 16 MHz. In this mode it was most common to center this band at 325 kHz to obtain spectra with fine structures





of SKR. However, in the frequency range of SKR also center frequencies at 125, 175, 225, 275, 525, and 1025 kHz were used. The frequencies from 125–275 kHz were mostly applied at periapsis passes later in the mission to obtain SKR spectra close to the local electron cyclotron frequency. No SKR was detected at the rarely used frequencies of 525 and 1025 kHz. The WBR usage of down-converted bands from HFR frequencies above 1–2 MHz had the intention to obtain high-resolution radio signatures of Saturn lightning, which are not the topic of this paper.

The WBR used only a single sensor to provide high-resolution electric or magnetic field measurements. It could either use a single electric monopole antenna ($E_u$, $Ev$, $E_w$) or the dipole antenna ($E_x$), and the latter measured the voltage difference between the $E_u$ and $E_v$ antenna (Gurnett et al., 2004). Since the dipole was more sensitive and less prone to spacecraft interferences than each monopole, the dipole antenna was almost exclusively used. Furthermore, it was also possible to connect the x-component of the magnetic search coil ($B_x$) or the Langmuir Probe to the WBR. The WBR had an instantaneous dynamic range of 48 dB. An automatic gain control was used to amplify the signal to a proper level in steps of 10 dB over a range of 0–70 dB, and the most commonly used gain was 40 dB.

The output of the chosen bandpass filter was sent to an 8-bit analog-to-digital converter with a sampling rate of 27.777 kHz for the 10-kHz wideband, and a sampling rate of 222.222 kHz for the 75-kHz wideband. The latter corresponds to a sampling time of $4.5\ \mu$s for each data point, and 1024, 2048, or 4096 samples were taken. For example, for 2048 data points a waveform series with a duration of $\sim 9.2$ ms was obtained which was losslessly compressed on board to minimize the data volume. The WBR could capture one waveform series once per multiple of 125 ms. It was most common to record one series every 125 ms, which constitutes the best temporal resolution of the WBR spectra. Sometimes a larger cycle time was used, e.g. the WBR made one snapshot each $5 \times 125 = 625$ ms at the second Venus flyby in June 1999, or each $3 \times 125 = 375$ ms at the Earth flyby in August 1999. The capture of 2048 data points of 8 bits every 125 ms corresponds to a maximum data rate of $\sim 131$ kbps (kilobit per second). As the allocated data rate for RPWS was much less than that, the WBR could of course only operate at selected times and not continuously. Operation time was usually on the order of minutes, but at selected times it could also be several hours. For operation in the baseband mode a typical duration was 1.5 minutes, whereas most WBR spectra centered at 325 kHz have durations of $\sim 35$ s.

The waveform series are Fourier transformed on the ground to build time–frequency spectrograms. For the 75-kHz wideband with a sampling frequency $f_S = 222.222$ kHz and $N = 2048$ data points the frequency resolution of the spectra is given by $f_S/N \approx 0.11$ kHz. The 25-kHz band from the HFR is sampled by the WBR in the 75-kHz mode, and thus the spectra at higher frequencies (e.g., 325 kHz) have a similar frequency resolution as the 75-kHz wideband. In this paper the 10-kHz wideband data is not used, and we just use WBR data from the 75 kHz baseband and from a down-converted band from the HFR centered at 325 kHz.

## 3 Classification scheme

The most common data product of the RPWS instrument are so-called dynamic spectra as displayed in Figure 1 and in the following figures. They give the power spectral density of a radio wave as a function of time and frequency. Usually the abscissa





(or x–axis) gives the spacecraft event time, and the ordinate (or y–axis) gives the frequency in a logarithmic or linear scale (linear scale in kHz in Figure 1). The power spectral density of the radio wave at a certain time and frequency is color coded
with units of $V^2$ $Hz^{-1}$ (as in Figure 1). It is quite typical to use a rainbow color scheme from blue to red color covering the dynamic range of the observed wave activity.

In this way a dynamic spectrum is quite similar to a two-dimensional painting on paper. In general one can distinguish geometrical objects by their dimension with dots as 0-dimensional objects, lines as 1-dimensional objects, and areas as 2-dimensional objects. This is the simple basis of our classification scheme. Our first class are the dots (DOTS). As linear
emissions should give us an idea about the movement of sources (Gurnett et al., 1979) we introduce four different classes for linear features: There are horizontal lines of constant frequency (HORZ), vertical lines at a fixed time (VERT), lines with a decreasing frequency with time having a negative slope (NEGS), and lines with an increasing frequency with time having a positive slope (POSS). The sixth class are simply areal features (AREA) with a certain extension in time and frequency. The 7th class are special cases (SPEC), which can be adapted for auroral radio emissions from various planets. For Saturn's SKR
we introduce the special cases of striations, rain, caterpillar, and worms. The terms 'striations' and 'rain' have been used before for AKR and SKR (Menietti et al., 2000; Menietti and Kurth, 2006), but the sub-classes of 'caterpillar' and 'worms' are newly introduced in this paper. Similar to the common term 'zebra pattern' for solar or Jovian radio emissions, we used catchy names of animals to describe the appearance of the features in the dynamic spectra. 'Zebra patterns' were first found in type IV solar radio emissions by Slottje (1972), but we do not use it here for SKR, because we found none. We briefly note that Kurth et al.
(2001) detected a peculiar feature of zebra patterns in Jovian broad-band kilometric radiation with the Cassini WBR.

Our classification scheme for SKR is summarized in Table 1, and it can be seen that two classes (DOTS and AREA) have no sub-classes. The three classes for linear features (HORZ, NEGS, POSS) can either be narrowbanded (HORZn, NEGSn, POSSn) or widebanded (HORZw, NEGSw, POSSw). Only vertical linear features (VERT) are intrinsically widebanded, and therefore they can be either short (VERTs) or long (VERTl) in duration. So all the linear features have two sub-classes, and
the 7th class of special structures (SPEC) consists of the four sub-classes mentioned above. In principle it would be possible to introduce more sub-classes, but we restricted ourselves to the most common ones.

Two examples for the classes of DOTS and AREA can be found in Figure 1 in the left and right panel, respectively. Several dots can be seen in the left spectrum mainly above 65 kHz, and we require in Table 1 that dots should have a bandwidth smaller than 2 kHz and a duration of less than 2 seconds. The narrowbanded drifting tones occurring every 15–20 kHz are spacecraft
interferences, and an inspection of the related browse plot at lower frequencies revealed that the strong emissions below 2 kHz with related vertical extensions are a mixture of spacecraft interferences and unresolved natural radio emissions. The two horizontal bands in the range of 20–25 kHz are not SKR, but Saturn narrowband emissions. We checked this by looking at low-resolution polarization and intensity spectra from the HFR (not shown), in which we found that the ordinary mode NB emissions are right-handed polarized, whereas in this case the extraordinary mode SKR comes from the southern hemisphere
and is left-handed polarized for Cassini being located at a latitude around 20°S. It is interesting to note that around the time of the WBR spectrum SKR shows a weak circular polarization above $\sim 50$ kHz, and an intensity that is close to the background and barely detectable. It is therefore conceivable that dots are not an own structure by themselves, but that other SKR structures





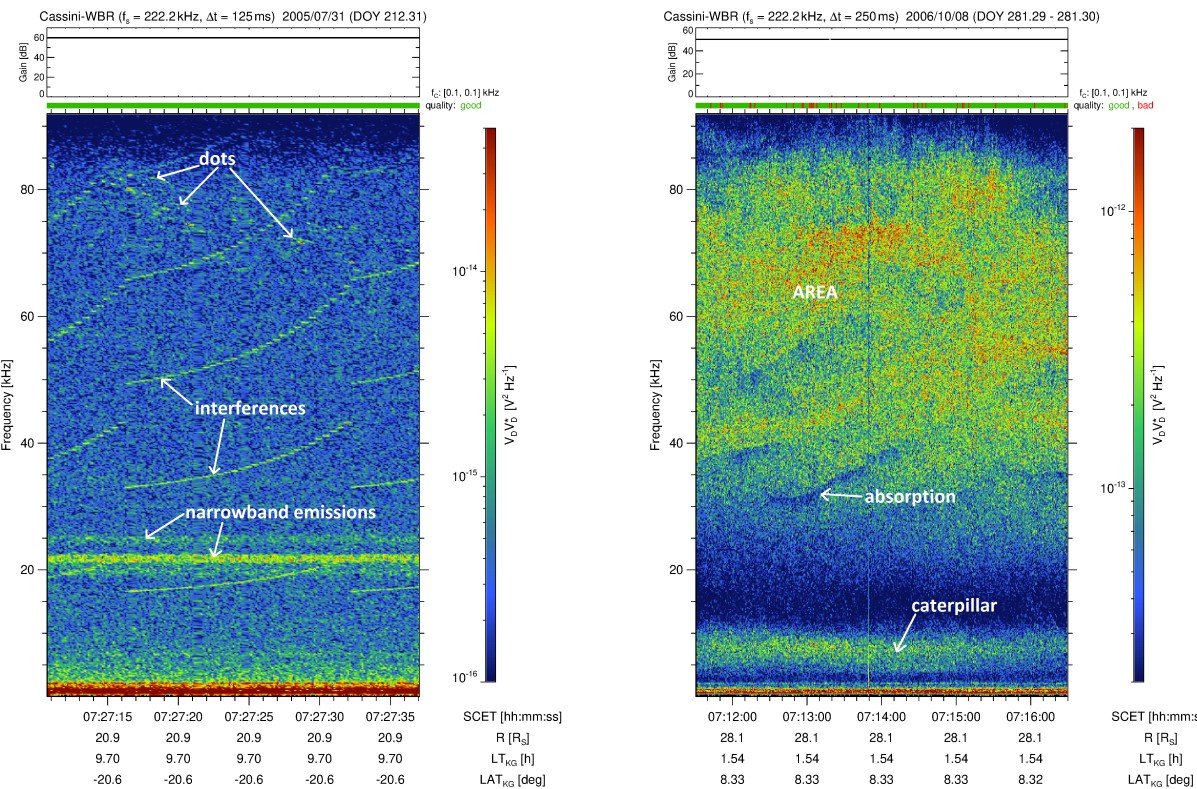

**Figure 1.** Fine structures of the classes DOTS (left side) and AREA (right side) reveal dot-like and patchy features of Saturn kilometric radiation, respectively. These Cassini WBR spectra show the color-coded auto-power spectral density of the radio waves received by the electric dipole antenna as a function of time (x–axis, in hh:mm:ss) and frequency (y–axis, in kHz). The numbers at the bottom denote the spacecraft event time (SCET), the distance of Cassini to Saturn's center in Saturn radii $R_S$, the local time (LT) of the spacecraft in hours, and its latitude (LAT) in kronographic (KG) coordinates in degrees. At the top one can find the sampling frequency $f_S$, the time resolution $\Delta t$, the date of the measurement, and additionally the day of year (DOY) is given in brackets. Furthermore, there is a panel indicating the gain state of the WBR in steps of 10 dB, and a green (or partly red) bar indicating the quality of the data. The color bar on the right gives the auto-power spectral density in $V^2\,Hz^{-1}$

.





**Table 1.** Summary table for seven different classes of SKR fine structures: The first column contains the number of the class, and the second column contains its name in a four-letter abbreviation. The third column denotes sub-classes: Vertical lines (VERT) can be short (VERTs) or long (VERTl) in duration. Horizontal lines (HORZ), and lines with negative (NEGS) or positive (POSS) slope can be narrowbanded (n) or widebanded (w). The forth column contains a detailed characteristic of the features in mathematical terms where $\Delta f$ denotes the bandwidth, $\Delta t$ denotes the time duration, $f_e$ and $f_s$ are the end and start frequency, and $t_e$ and $t_s$ are the end and start time, respectively. For frequencies the unit is kHz, and for times the unit is seconds (only indicated for DOTS class). Finally, the fifth and last column contains a verbal description for each sub-class.

| No. | Class name | Sub-class | Characteristics and criteria | Verbal description |
|---|---|---|---|---|
| 1 | DOTS | - | $(\Delta t \leq 2 \text{ s}) \wedge (\Delta f \leq 2 \text{ kHz})$ for each dot | dots |
| 2 | HORZ | HORZn | $(\Delta f \leq 1) \wedge (\Delta t > 10)$ | horizontal line, narrow |
| | | HORZw | $(1 < \Delta f < 5) \wedge (\Delta t > 10)$ | horizontal line, wide |
| 3 | VERT | VERTs | $(\Delta t \leq 1) \wedge (\Delta f > 10)$ | vertical line, short |
| | | VERTl | $(1 < \Delta t < 5) \wedge (\Delta f > 10)$ | vertical line, long |
| 4 | NEGS | NEGSn | $(\Delta f \leq 1) \wedge [(|f_e - f_s| > 10) \vee (t_e - t_s > 10)]$ | line with negative slope, narrow |
| | | NEGSw | $(1 < \Delta f < 5) \wedge [(|f_e - f_s| > 10) \vee (t_e - t_s > 10)]$ | line with negative slope, wide |
| 5 | POSS | POSSn | $(\Delta f \leq 1) \wedge [(f_e - f_s > 10) \vee (t_e - t_s > 10)]$ | line with positive slope, narrow |
| | | POSSw | $(1 < \Delta f < 5) \wedge [(f_e - f_s > 10) \vee (t_e - t_s > 10)]$ | line with positive slope, wide |
| 6 | AREA | - | $(\Delta t > 10) \wedge (\Delta f > 10)$ | areal feature |
| 7 | SPEC | striations | $\Delta f \leq 1$ | bundle of many sloped and narrow lines |
| | | rain | $\Delta f \leq 1$ | short, interrupted striations |
| | | worms | $\Delta f \leq 5$ | wiggly, broader lines |
| | | caterpillar | $\Delta f \approx 10, \Delta t$ a few hours | coarse structure |

are hidden below the intensity and fluctuation of the receiver noise. We just cannot see these structures, and only the dots stick out from the background like mountain peaks above a fog layer. The right spectrum in Figure 1 is the opposite case to the

almost empty spectrum on the left: It shows a large area of SKR emissions from $\sim 25$ kHz up to more than $\sim 80$ kHz. Our requirement for the structure AREA is that it should have a bandwidth of at least 10 kHz and a duration of at least 10 s, and this is clearly fulfilled here. The fall-off in intensity at the higher frequencies is caused by the frequency response curve of the 75-kHz WBR filter (see Figure 22 of Gurnett et al. (2004)). The fall-off of the intensity at 80 kHz is not very significant, so that we can assume a frequency bandwidth of at least 80 kHz for the nominal 75-kHz WBR, and in the following we will

name it 80-kHz WBR. Similarly, the down-converted data from the HFR show that the bandwidth at higher frequencies is also somewhat larger than the nominal 25 kHz, and practically we can assume a bandwidth of 30 kHz around the center frequency of 325 kHz as the right spectrum in Figure 2 is demonstrating. The emission band named 'caterpillar' from about 3 to 11 kHz in the right panel of Figure 1 will be discussed later.





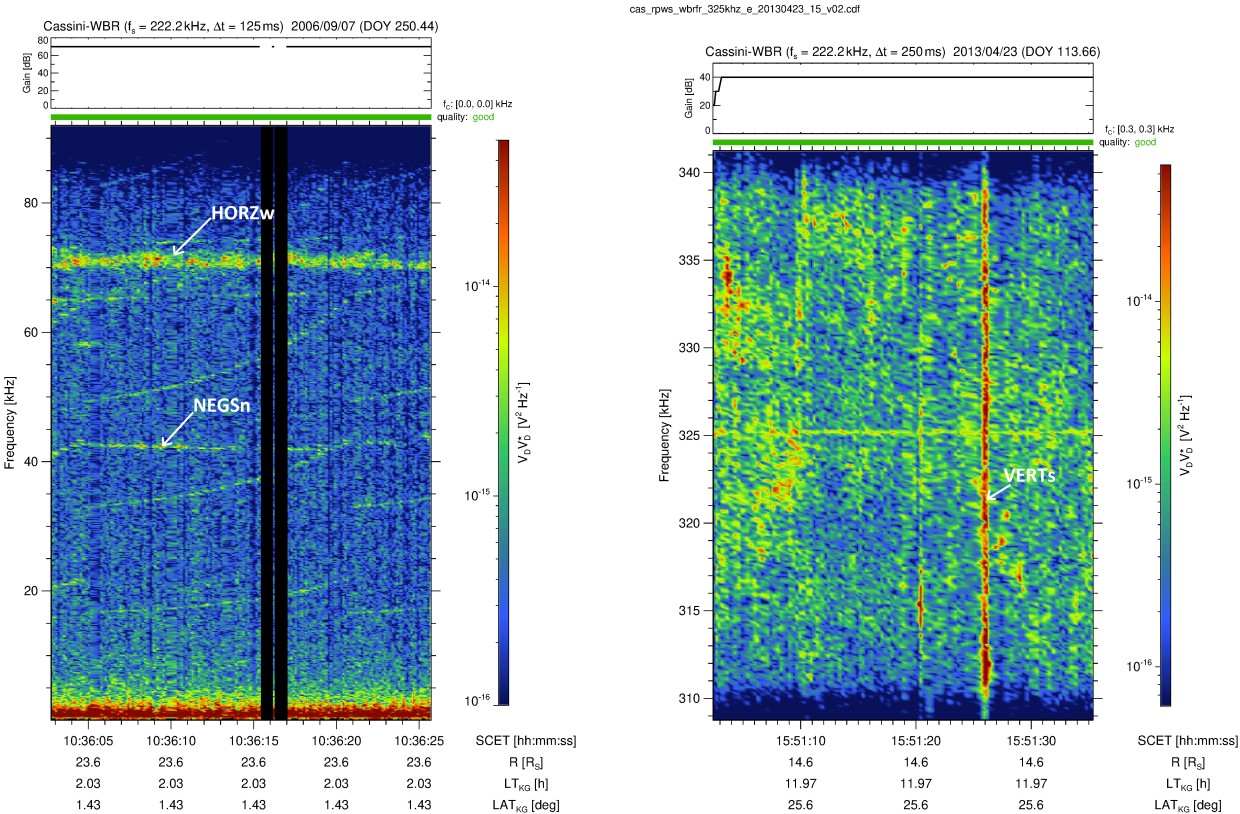

**Figure 2.** Fine structures of the classes HORZ (left side) and VERT (right side) show horizontal and vertical linear features of Saturn kilometric radiation, respectively.

The left spectrum of Figure 2 shows a horizontal line with a bandwidth of $\Delta f \approx 3$ kHz right above 70 kHz. It lasts from

the beginning until the end of the spectrogram, and so it has a duration of about 23 s. Its practically constant frequency and bandwidth qualifies it as a widebanded, horizontal emission (HORZw). We only record the most prominent emissions, and therefore we require a minimum duration of 10 s for all horizontal emissions as noted in Table 1. Another requirement to be fulfilled for horizontal emissions is a bandwidth of $\Delta f < 5$ kHz, because it should still rather look like a linear than a patchy structure. We also note that we checked the low-resolution HFR spectrum, and the polarization indicated that the horizontal

emissions should be SKR and not a Saturn narrowband emission. Similar to the left side of Figure 1 the group of thin positively drifting lines and the emissions below 2 kHz are mostly spacecraft interferences. However, there is also one thin line with a very small negative drift as indicated in the figure. This feature should be classified as a narrowbanded, negatively drifting line (NEGSn). The right spectrum of Figure 2 has a center frequency of 325 kHz, and it is not in the 80-kHz baseband. The thin horizontal line at the center frequency of the passband at exactly 325 kHz is probably due to the mixing process for the

down-conversion. One can clearly see a strong vertical line at 15:51:26 which lasts about a second, so it qualifies as a short





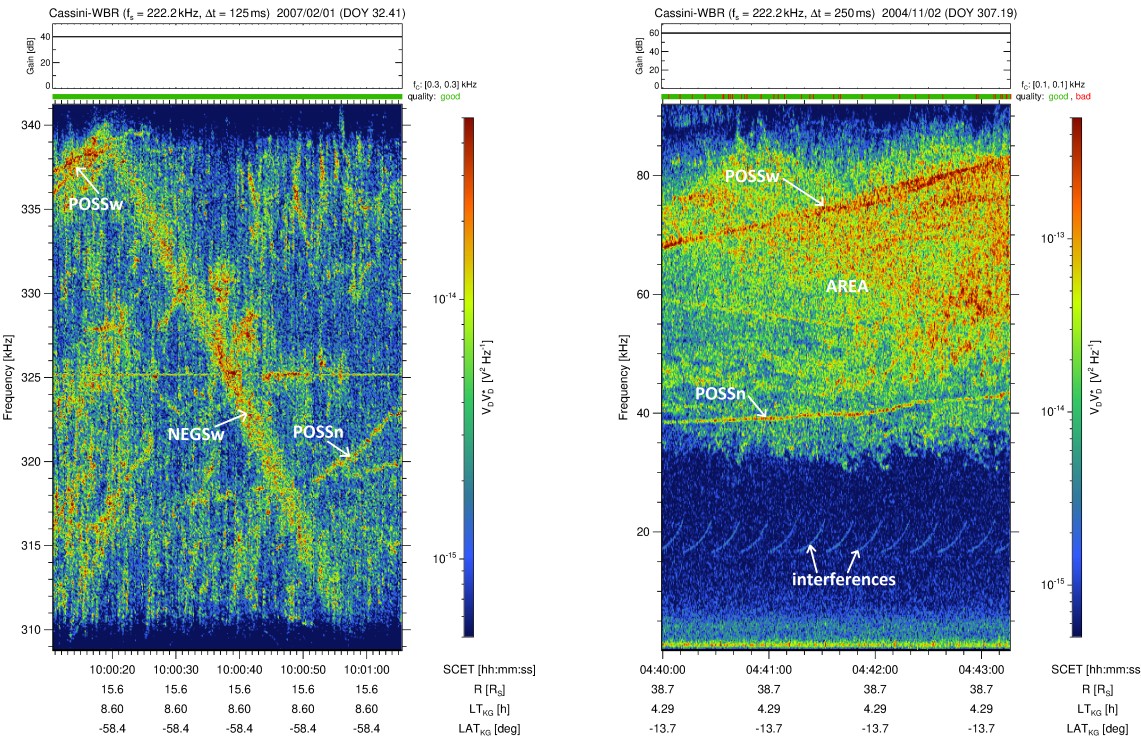

**Figure 3.** Fine structures of the classes NEGS (left side) and POSS (right side) show linear features of Saturn kilometric radiation with negative or positive slope, respectively.

vertical streak (VERTs). There seems to be a second vertical streak around 15:51:20, but it is not strong enough to be a vertical structure (VERT). It only has a large intensity around $(315 \pm 2)$ kHz, but we require a strong intensity of at least several dB above background over a bandwidth of at least 10 kHz (see Table 1). The irregular shaped emissions on the left side are too patchy and interrupted to qualify as an AREA. Note that for most SKR spectra the background intensity is typically in the

range of $10^{-15}$ V$^2$ Hz$^{-1}$ to $10^{-16}$ V$^2$ Hz$^{-1}$, where it is rather at the higher end around 325 kHz and at the lower end below 80 kHz. At such frequencies the background noise is often dominated by instrumental noise, but additional noise can come from thermal electrons.

The spectrum on the left side of Figure 3 was also taken around 325 kHz, and it also has a horizontal spacecraft interference at its central frequency. The left spectrum should illustrate a line with a negative slope, and one can easily see this rather

widebanded line all across the spectrum from the upper left to the lower right, which clearly classifies as a NEGSw structure. However, one can also see at least two lines with a positive slope in the left spectrum of Figure 3, and both last about 10 seconds or a little longer. The first one is in the beginning of the spectrum above 335 kHz, and it connects to the upper left corner of the large NEGSw line. This line is clearly widebanded and so it classifies as POSSw, whereas the second positive line is narrowbanded (POSSn). It can be seen during the last 13 s of the spectrum around 320 kHz. We note that for both positive





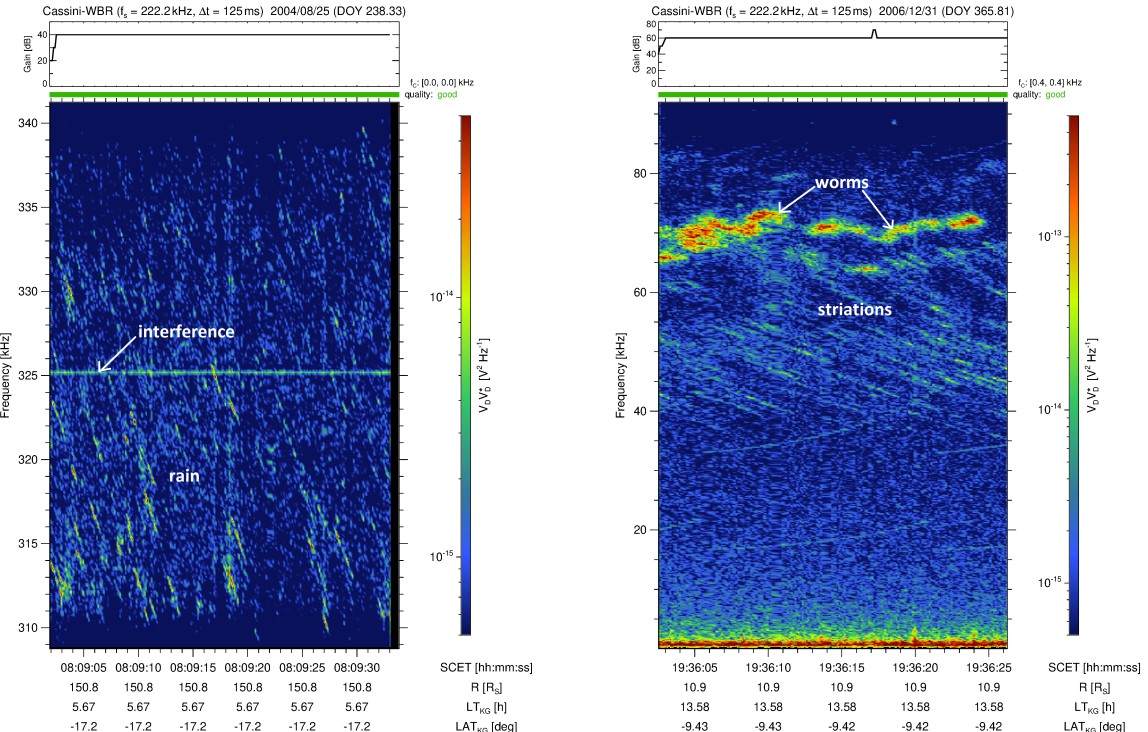

**Figure 4.** SKR fine structures called 'rain' at 325 kHz in comparison to 'striations' at 80 kHz measured by the Cassini WBR. Above 60 kHz the newly defined features called worms can be seen.

(POSS) and negative lines (NEGS) our criteria in Table 1 require either a duration of at least 10 s or an extent in frequency of 10 kHz, where the frequency extent should not be mixed up with the instantaneous bandwidth $\Delta f$ (they are only the same for a vertical line). Lines with a small negative or positive slope last longer and should rather fulfill the criterium for the duration, whereas lines with large slopes have a large extent in frequency. The right spectrum in Figure 3 shows one widebanded line with positive slope (POSSw) and a narrowbanded one (POSSn), but for this spectrum the lines are also embedded in an areal

feature (AREA).

We now come to the 7th class of special cases (SPEC). In Figure 4 we compare the 'rain' feature around 310–330 kHz on the left side with the 'striations' feature around 35–65 kHz on the right side. Both sub-classes consist of a multitude of narrowbanded lines with a negative slope. The major difference is that the lines of 'rain' are usually shorter than the lines of 'striations'. The frequency extent of the former is just a few kHz, whereas the extent of the latter is typically some tens

of kHz. We note that such discrete, negative–slope features extending over a period of several seconds were initially named 'stripes', and they were seen in AKR data of the spacecraft DE-1, Polar, and Galileo (Menietti et al., 1996, 1997). Later the terms 'striations' and 'rain' were introduced for these features and used as synonyms (Menietti et al., 2000; Menietti and Kurth, 2006). Finally the term 'striations' or 'striated AKR' has prevailed and is most commonly used (Mutel et al., 2006; Menietti





et al., 2006). Here we want to re-introduce the term 'rain', but use it exclusively for short, interrupted striations with a frequency
extent of just a few kHz. We assume that 'striations' and 'rain' are essentially based on the same physical mechanism, but we
consider it as an important structural difference if they are largely continuous over tens of kHz or interrupted. Looking closely
at the spectrum of 'striations' on the right side of Figure 4 one can also see some isolated 'raindrops', but the overall tendency
of the features extending over tens of kHz in contrast to the 'rain' spectrum on the left side is very clear. Another example
would be Figure 1 in Menietti et al. (2006) showing rain, whereas Figure 4 in Mutel et al. (2006) clearly shows striations.
We note that striations with a positive slope also exist for SKR, and such a rare example is shown later in the right panel of
Figure 5. For AKR, Menietti et al. (2000) stated that the vast majority of striations have negative drift rates between $-8$ and
$-2\,\mathrm{kHz\,s^{-1}}$, and that only a much smaller group of striations with positive drift rates exists.

Finally, the right panel of Figure 4 shows a spectral feature which is newly introduced in this paper, and it can be seen
mainly between 60 and 75 kHz. It consists of wiggly lines with a bandwidth of a few kHz that slightly move up and down
in frequency. Like striations, these features tend to occur in groups and we named them 'worms', because they look like a
group of wriggly worms in freshly turned garden soil. Some of them only last for a few seconds, while others can also be tens
of seconds long. The second newly introduced feature nicknamed 'caterpillar' can be seen in the right spectrum of Figure 1.
Below the large areal feature from $\sim 25$–80 kHz there is a band of constant frequency in the range of $\sim 3$–11 kHz being
present for about 5 minutes, which is the whole recording time of the spectrum. Indeed, using HFR low-resolution spectra (not
shown) we found that these 'caterpillar' features usually last for several hours and do not change in frequency. So they are
rather a coarse structure of SKR at lower frequencies, and their central frequencies were found ranging from a few kHz up to
$\sim 40$ kHz. However, the WBR usually shows a smooth spectrum for caterpillars, without particular fine structure, except an
occasional increase of intensity near the central frequency of the emission. Their typical bandwidth $\Delta f$ is $\sim 10$ kHz, which
makes them easily distinguishable from Saturn narrowband emissions (Ye et al., 2009) in the 80-kHz WBR spectra. As can be
seen in the left panel of Figure 1 at 20 kHz, the narrowband emissions have a smaller bandwidth of just a few kHz. Similarly,
also narrowband emissions at 5 kHz have a small bandwidth of just $\sim 2$ kHz as the left panel of Figure 5 shows, and their
complex fine structure can only be seen in 10-kHz WBR data (see Figure 4 of Wang et al. (2010)). We do believe that the
'caterpillars' are a special part of low-frequency SKR, as low-resolution polarization spectra (not shown) reveal that their
polarization characteristics are rather similar to SKR. Therefore, we do include caterpillars as special fine structures of SKR,
and we plan to investigate them in more detail in a future paper.

## 4 Statistics of fine structure occurrence

We used our classification scheme of Table 1 to classify more than 5500 spectra at low frequencies in the 80-kHz baseband, as
well as almost 4000 spectra at medium frequencies around 325 kHz. In the previous section we have presented examples of all
classes and sub-classes, and we did not classify very small features, but only the most prominent and large ones which fulfilled
the criteria set down in the 4th column of Table 1. We rather tended to dismiss features which were unclear or too weak.





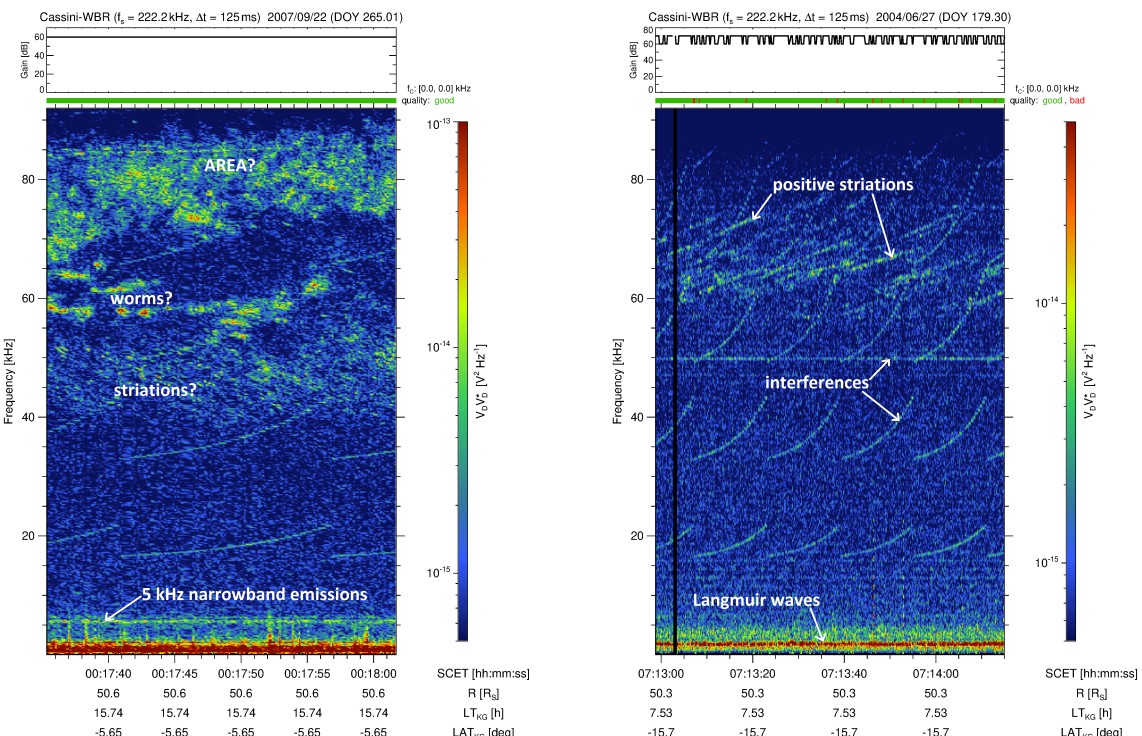

**Figure 5.** Example of unclassified SKR spectrum on the left side, and rare positive striations on the right side.

Some spectra might not have any classifiable structures in them, and an example for such a spectrum is shown in the left panel of Figure 5. Here some worms might be present around 60 kHz, but their structure is unclear as they do not really show wiggly features. This is why we dismissed them as pointed out above. There might be some striations between 40 and 50 kHz, but their structure is not really evident. Finally, the patchy features above ∼ 70 kHz were also not classified as an AREA

because of frequent perforations by background noise. Therefore, we consider this SKR spectrum as unclassified (UNCL). We note that several spectra also have unclassified structures next to classified ones. An example for this is the left spectrum of Figure 3 which contains some classified lines, but also lots of other unclassified emissions. We do not count these, but we only count the number of spectra without any classified linear, areal or special structures. The class of DOTS is also special in the way it is counted. Little dots are very common features, and one can find them in many spectra. For example, besides the DOTS

spectrum in Figure 1, one can see some dots around 65 kHz in the left spectrum of Figure 2, and several dots around the linear features in the left spectrum of Figure 3. There are also several dots around the 'raindrops' in the left spectrum of Figure 4. All of these dots are not counted, and we just count the number of spectra that contain only dots and no other SKR fine structures. So our class of DOTS indeed means DOTS ONLY.





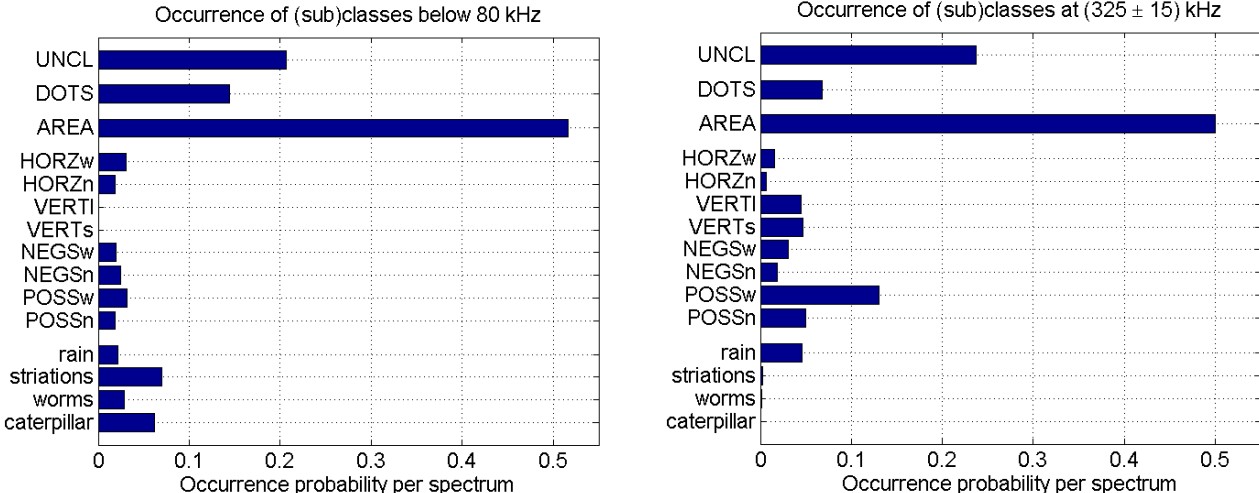

**Figure 6.** Two bar charts showing the occurrence probabilities per spectrum of various classes/sub-classes of SKR fine structures for lower frequencies (below 80 kHz) on the left side and for medium frequencies (around 325 kHz) on the right side. The statistics are based on a total number of 5551 SKR 80 kHz WBR spectra and on 3973 WBR spectra at 325 kHz, respectively.

The linear features of the classes HORZ, VERT, NEGS, POSS are counted very differently to the dots. In our first statistic

we count the number of spectra that contain a certain linear class. We note that we count only one AREA per spectrum when it is present, i.e. here we also count the number of spectra containing the class AREA. In the same way we proceed with the sub-classes of SPEC. The sub-classes of rain, striations and worms are group phenomena, and we neither count the number of raindrops, single striations or worms, nor if there are multiple groups of them in one spectrum. A caterpillar seems to be a unique phenomenon, and we only found just one caterpillar in one spectrum and not multiple ones at different frequencies.

Similar to AREA, we only count the number of spectra containing the sub-classes of rain, striations, worms, and caterpillar. Additionally, it is also obvious that some spectra can have structures of multiple classifications. For example, the right spectrum of Figure 4 contains the sub-classes striations and worms, and the right spectrum of Figure 3 has an AREA with embedded lines of positive slope (POSSw and POSSn). Therefore, in Figure 6 the sum of all occurrence probabilities does not add up to one, but it is larger than one. Each fractional number can be seen as the occurrence probability for a certain class or sub-class

to occur in an average WBR time–frequency spectrogram. It is computed as the number of spectra containing a certain class, divided by the total number of investigated spectra, and for the latter we only take spectra containing SKR.

The left panel of Figure 6 shows that about 21% of the 80-kHz WBR spectra had no classifiable SKR structures, whereas for the 325-kHz spectra this percentage is a bit higher with $\sim 24\%$, as the right panel shows. For low frequencies $\sim 14\%$ of the spectra had dots only, whereas for medium frequencies it was $\sim 7\%$. Areal features have a similar occurrence probability

around 50% for both low and medium frequencies. However, we point out that such a comparison as well as the absolute values have their limitations, because they highly depend on the extension of the time–frequency plane of the spectrum. The more we enlarge the size of the spectrum, the more it is likely to find a classifiable structure like an AREA, linear, or special features. If





the size of the spectrum is large enough, we might find an areal object (AREA) in almost every spectrum, thereby pushing the occurrence probability close to one.

As the size of the time–frequency plane is different for the 80-kHz spectra compared to the $(325 \pm 15)$ kHz spectra, it is advisable to rather compare the features of the 80-kHz spectra among each other, and to do a separate internal comparison of features of the 325-kHz spectra: For the linear features below 80 kHz the left panel of Figure 6 shows that they all have a similar occurrence probability around 2-3% except for the class of vertical lines (VERT), where it is just $\sim 0.05\%$. We again note that in this statistic it does not matter if there are 1, 2 or 3 lines of the same sub-class in one spectrum. Hence, this number

tells you the probability to find a spectrum with at least one line of a certain sub-class in the spectrum. The special sub-classes of rain and worms also have an occurrence probability of 2–3%, but striations and caterpillars are much more common with a probability of $\sim 6 - 7\%$. We note that below 80 kHz the continuous lines of striations are about three times more common than the interrupted lines of rain.

       Now let's look separately at the statistics of the SKR fine structures around 325 kHz in the right panel of Figure 6: Here the

probability of vertical lines (VERT) is around 4.5% for both sub-classes, whereas it is harder to find horizontal lines (HORZ) which occur with probabilities of 1.6% for HORZw and 0.7% for HORZn. Lines with negative slope (NEGS) are again easier to find if widebanded ($\sim 3\%$, NEGSw) compared to narrowbanded ($\sim 1.8\%$, NEGSn). Surprisingly, the lines with positive slopes (POSS) are much more common with about 5% for POSSn and 13% for POSSw. For the special class (SPEC) the sub-class of rain is more than one order of magnitude as common ($\sim 4.6\%$) as striations ($\sim 0.3\%$). We only found 6 spectra with

worm like features at 325 kHz (0.15%) and no caterpillar, so the latter are probably completely absent at medium frequencies.

       In the second statistic we count the number of the lines directly. For example, we found that the left spectrum of Figure 3 contains 1 NEGSw, 1 POSSw, and 1 POSSn, and that the right spectrum of the same figure contains 1 POSSw and 1 POSSn embedded in an AREA. The left spectrum of Figure 2 contains 1 HORZw and 1 NEGSn, and the right one of the same figure has just 1 VERTs. Furthermore, we do not count these numbers per spectrum, but per spectral area in the SKR frequency

range. For this we evaluate the size of the time–frequency plane by multiplying the duration of the spectrum with its frequency extent, and this area is different for the average 325 kHz spectrum compared to the average 80 kHz spectrum. For the 325 kHz spectra the frequency extent is always $\sim 30$ kHz. Their time duration was mostly 35 s and sometimes 32 s, and we calculated an average duration of $(34.8 \pm 0.8)$ s. Hence the spectral area is $A_{325kHz} = 30 \cdot 10^3 \cdot 34.8 = 1.04 \cdot 10^6$ Hz s $\approx 1$ MHz s. The duration and frequency extent for a typical 80 kHz spectrum is much more variable, and we found an average duration of

$(96 \pm 23)$ s. The 80-kHz spectra typically lasted 52, 80, 90, or 105 s, but sometimes also much longer (up to over one hour) and such spectra lasting longer than 4–5 minutes were excluded from our classification. For the frequency extent we simply cannot take the whole 80 kHz as this is the frequency region where the low frequency cutoff of the SKR can be found. Note that the 325-kHz spectra are fully within the typical SKR frequency range, but that is not the case for the 80-kHz spectra. Therefore, we determined the average SKR cutoff frequency from 100 arbitrarily chosen 80-kHz spectra among those five thousand which

we classified, and we found a value of $(40 \pm 19)$ kHz. So the average SKR spectral area for the 80-kHz spectra is given as $A_{80kHz} = (80 - 40) \cdot 10^3 \cdot 96 = 3.84 \cdot 10^6$ Hz s $\approx 4$ MHz s. This is about four times larger than for the 325-kHz spectra. Taking the spectral area into account should now allow us to make a better comparison between low and medium frequencies.





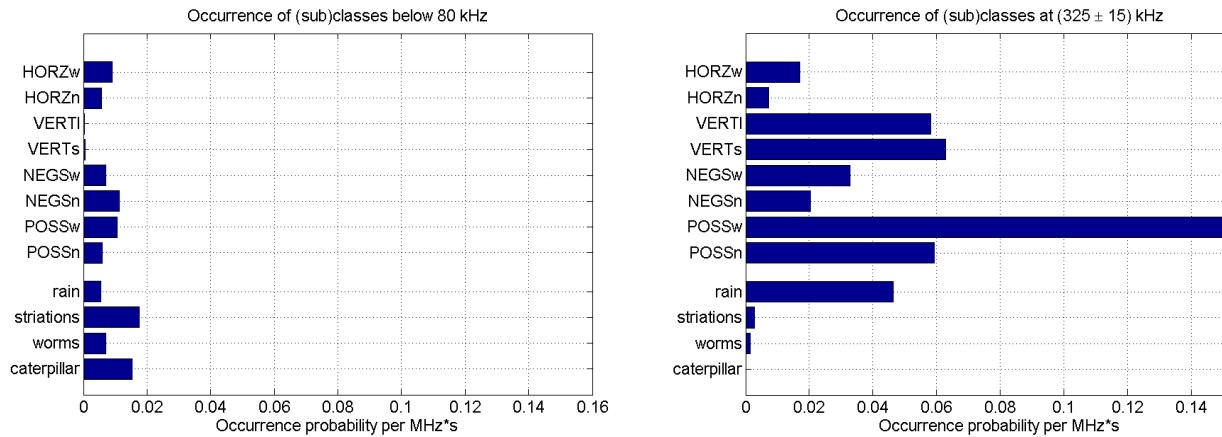

**Figure 7.** Two bar charts showing the occurrence probabilities per spectral area of 1 MHz·s of linear and special SKR fine structures for lower frequencies (below 80 kHz) on the left side and for medium frequencies (around 325 kHz) on the right side.

Figure 7 shows the occurrence of linear and special SKR fine structures per spectral area of 1 MHz·s. The classes UNCL, DOTS, and AREA are not plotted since they largely depend on the actual extension of the spectral area (as pointed out above), whereas linear (HORZ, VERT, NEGS, POSS) and special (rain, striations, worms, caterpillar) structures can be counted more easily. The occurrence probability of a certain subclass (e.g., HORZn) is now given as the total number of lines (e.g., narrowbanded horizontal lines) divided by the total spectral extension of all investigated spectra in MHz·s. The right plot of Figure 7 for 325 kHz is very similar to the right plot of the previous bar chart in Figure 6, only the occurrence probabilities are somewhat higher, since now we count the number of lines and not the number of spectra containing lines. Again, the occurrence for widebanded lines with a positive slope (POSSw) is very prominent and high with more than 0.15 lines per spectral area of 1 MHz·s. The left plot of Figure 7 shows that the occurrences of linear features is much less for low frequencies below 80 kHz with occurrences generally below 0.02 lines per 1 MHz·s. Similar to before, vertical lines (VERT) are almost absent below 80 kHz. Striations are more common at low frequencies with an occurrence probability of 1.8% per MHz·s below 80 kHz compared to 0.28% per MHz·s at medium frequencies around 325 kHz. The sub-class of rain is still more common at medium frequencies with 4.6% compared to 0.5% at low frequencies. Thus it seems that the lines of striations tend to get interrupted and appear as rain at medium frequencies.

As mentioned in the beginning of this section, we rather tended to dismiss weak and unclear features, and therefore our occurrence probabilities can be seen as lower limits. There is also a certain subjectivity in our classification, as the discussion of the unclassified features in the left spectrum of Figure 5 has demonstrated. We estimate that the error of the various occurrence probabilities in Figures 6 and 7 is around 10-20%. For example, it can hardly be said if horizontal (HORZ), negatively drifting (NEGS) or positively drifting (POSS) lines are more common below 80 kHz. However, our main results from this section, namely that the vertical lines are almost absent below 80 kHz, that the widebanded lines with a positive slope (POSSw) have a much higher occurrence probability than other lines at 325 kHz, that striations at low frequencies rather turn to rain at medium





frequencies, and that worms and caterpillars are almost exclusively present below 80 kHz are very solid statistical results. And
it will be just these statistical results which are discussed in the next section.

## 5 Discussion

We now discuss physical implications of various SKR fine structure classes. We found that $\sim 10\%$ of the spectra show only
dots (DOTS) and no other structures, but that dots are practically present in many SKR WBR spectra. A single dot can be
interpreted as a single radio source at a fixed altitude on a fixed magnetic field line, which is active for a short time ($< 2$ s).
On the other hand, a dot can be a single peak of an unknown structure hidden by the receiver noise, and we cannot really
distinguish these two cases. On the contrary, the class of AREA can be interpreted as consisting of many dots, which means
there should be many radio sources at different altitudes radiating at the same time. Furthermore, it is probably necessary
that not only one magnetic field line is involved in this, but many. It is well known that auroral radio emissions and their
corresponding footprints in ultraviolet (UV) light usually occur at relatively fixed latitudes, but stretch over a large range of
longitudes or local times. For Saturn, the relation between SKR and UV footprints has been shown by Lamy et al. (2009), and
the UV aurora is usually located between $70°$ and $80°$ latitude (Badman et al., 2006; Nichols et al., 2016), but can stretch over
several hours of local time. One of the difficulties with SKR is that it is a beamed radio emission with an average beaming
angle of around $70°$ with respect to the magnetic field direction at the source. Direction-finding measurements found that the
range for the beaming angle is large and goes from $\sim 30°$ up to $90°$ with a probable dependence on frequency (Cecconi et al.,
2009; Lamy et al., 2011). Wave growth rate calculations for SKR by Mutel et al. (2010) suggest that significant wave growth
can only be achieved close to perpendicular propagation ($90° \pm 10°$), so that refraction of radio waves close to the source must
take place to arrive at the observed beaming angles. The beaming leads to a reduced visibility of sources, and a specific SKR
source at a certain altitude and magnetic field line can only be observed by a single spacecraft like Cassini if it is located within
the emission beam. However, planetary radio emissions simulations by Lamy et al. (2013) show that active auroral source
regions 6 hours wide in local time ($90°$ in longitude) would have enough visible SKR sources to create an areal structure in the
dynamic spectrum observed by Cassini (see their Figure 11). In general, at low temporal resolution the coarse structure of SKR
shown in most RPWS spectra is one of areal features lasting for several hours. Regions devoid of SKR within the areal features
can be caused by non-active field lines or by emission beams missing the spacecraft. It is therefore not surprising that areal
structures as defined in our Table 1 are also present in about half of all SKR high temporal resolution spectra. Some authors
suggested that the whole SKR emission including areal features consists of many small and mostly linear fine structures, which
until today have not been fully resolved (e.g. Pottelette et al. (2001); Mutel et al. (2006); Treumann (2006)). This assumption is
based on the fact that the CMI mechanism is thought of being intrinsically narrowbanded (McKean and Winglee, 1991; Yoon
and Weatherwax, 1998).

We note that there is a limit for the time and frequency resolution in signal processing. This is the so-called Gabor limit,
which results from the application of Heisenberg's uncertainty principle to time–frequency analysis (Gabor, 1946), and it is
given by the relation $\Delta f_{eff} \Delta t_{eff} \geq \frac{1}{2}$ with $\Delta f_{eff}$ and $\Delta t_{eff}$ as the so-called effective frequency and duration, respectively.





For a single WBR snapshot of 2048 points, which is processed on the ground, we find a frequency resolution of $\Delta f = 0.11$ kHz and a time resolution of $\Delta t = 9.2$ ms (see Sect. 2), which leads to a product of $\Delta f \Delta t \approx 1$. Indeed, the value of this time–bandwidth product depends on the definition of $\Delta f$ and $\Delta t$, and in our case we must use Küpfmüller's uncertainty principle with $\Delta f \Delta t \geq 1$, which is exactly fulfilled. However, due to data rate limitations the WBR takes at best only one snapshot every 125 ms, which increases the product to 0.11 kHz × 125 ms≈ 14. Note that the frequency resolution can be improved at the expense of the time resolution, e.g. using a snapshot of 4096 data points yields a frequency resolution of $\sim 54$ Hz. Kurth et al. (2005) further improved the frequency resolution to 27 Hz by using 4 consecutive WBR snapshots of 2048 samples each to find the bandwidth of a linear feature in SKR fine structure.

Theoretical considerations suggest that the cyclotron maser can not only generate radio emissions, but also be responsible for their absorption. This takes place in regions where the electron distribution function $f$ has a negative slope with respect to the perpendicular velocity $v_\perp$, i.e. $\partial f / \partial v_\perp < 0$ (Treumann, 2006). More exactly, if the parts along the resonance ellipse in velocity space with $\partial f / \partial v_\perp < 0$ outweigh the parts with $\partial f / \partial v_\perp > 0$, the net growth of waves is negative, which means the resonant electrons gain energy and may experience pitch angle scattering and the waves are absorbed. An absorption signature should best be seen within areal features, but we found only a few of them in SKR spectra. The right panel of Figure 1 contains an example where a dark blue line with a positive slope can be seen starting around 07:12:30 SCET at 30 kHz. Another absorption feature is maybe present around 07:12:10 and 38 kHz. It is unclear why absorption features are so rare. Treumann (2006) pointed out in his Figure 30 that an emission from an electron hole consists of a combination of an emission and an absorption line with the absorption at the high-frequency side of the emissions. In case such an emission line is surrounded by receiver noise like in the left panel of Figure 2, one would not be able to see the corresponding absorption line. However, we also found many examples of strong linear features embedded within a large areal feature as in the right panel of Figure 3, and none of them was accompanied by an absorption line at the high-frequency side. Therefore, it is unlikely that SKR is created by electron cyclotron radiation from electron holes. Additionally, Mutel et al. (2007) found that electron holes cannot be elementary radiation structures as their perturbations do not enhance the growth rate.

It is straightforward to assume that linear features are caused by a radiating source moving upwards or downwards along a magnetic field line (Gurnett et al., 1979). Some sources may also stay at a constant altitude for several seconds, which would manifest itself as a horizontal line (HORZ). Lines with a negative drift rate (NEGS, decreasing frequency with time) should develop for upward moving sources, whereas lines with a positive drift rate (POSS, increasing frequency with time) have downward moving sources, moving towards regions with higher magnetic field strength. Vertical lines (VERT) must either consist of many sources at various altitudes radiating at the same time, or must be sources with a high negative or positive drift rate which cannot be resolved by the receiver. Vertical lines going over a range of 30 kHz around the center frequency of 325 kHz, as shown on the right side of Figure 2, should have a high positive or negative slope beyond $\pm 240$ kHz s$^{-1}$ as the time resolution of the WBR is typically 125 ms. Using equation (6) of Gurnett and Anderson (1981) and adapting it to Saturn with a simple dipole magnetic field model (magnetic moment of 0.2154 R$_S^3$ G yielding an electron cyclotron frequency of $f_{ce} = 1570$ kHz at Saturn's surface at $75°$ latitude) this drift can be translated to a parallel velocity along the magnetic field line of $\sim 25,100$ km s$^{-1}$ or about 8% of the vacuum speed of light. This is not totally unrealistic since 10 keV electrons have





a speed of $\sim 60,000$ km s$^{-1}$, and in case of a low pitch angle their parallel motion might be that fast. However, each discharge or current surge in one of the Cassini instruments can cause a vertical line in the RPWS receiver, and therefore it cannot be totally excluded that some or all the vertical lines found at 325 kHz are caused by spacecraft interferences. The gain plot in

the small upper panel on the right side of Figure 2 shows that the vertical lines are not caused by sudden gain changes in the WBR. Interestingly, below 80 kHz vertical lines are almost completely absent. There is an example with vertical lines around 60 kHz having a bandwidth of $\sim 20$ kHz. This corresponds to a slope of larger than $\pm 160$ kHz s$^{-1}$ and a source speed of 53% of the vacuum speed of light. This is clearly unrealistically high, and the rare vertical lines found below 80 kHz might be due to spacecraft interferences. The typical slope of striations or other linear features in SKR is in the range of a few kHz

per second, corresponding to speeds of a few hundred km s$^{-1}$ at medium frequencies (325 kHz) or a few thousand km s$^{-1}$ at low frequencies (<80 kHz). For AKR such speeds have been linked to electromagnetic ion cyclotron waves traveling along the auroral field lines and stimulating AKR emission (Menietti et al., 2006; Menietti and Kurth, 2006).

Concerning the occurrence of linear features at medium (MF) and low frequencies (LF), it is interesting to note that according to Figure 7 most of them are several times more common at medium frequencies compared to low frequencies. One reason for

this could be that the sources need to drift a shorter distance at MF compared to LF for the same frequency extent. This is due to the fact that the magnetic field decreases faster with increasing altitude close to the planet where the emissions at MF are generated. We also found that the occurrence of widebanded lines with a positive slope is very high at medium frequencies, and around 13% of the spectra show the class POSSw around 325 kHz (see Figure 6). This might be an indication that many SKR sources are due to unstable down-going electron populations, and so far the majority of crossed SKR source regions were

associated with magnetic field signatures that indicate an upward current region (Lamy et al., 2010; Schippers et al., 2011; Lamy et al., 2018). Lines of positive slope need down-going electron populations which remain CMI–unstable while they propagate over larger distances. This means they are not immediately converting all their free energy into wave amplification at one fixed location, i.e. at a single frequency, but the instability is so strong that it is gradually diminished and converted into wave energy while the population is propagating downward. Such unstable down-going electron populations seem to be more

common closer to the planet than further away where the low-frequency SKR is generated.

The special structures of rain and striations have already been broadly discussed at the description of Figure 4 in Sect. 3. Here we only add that the negative slopes of rain and striations in the left and right panel of Figure 4 are about the same and have a value of $\sim -2$ kHz s$^{-1}$, which is at the lower end (in magnitude) of drift rates for AKR striations, found to be in the range of $-8$ to $-2$ kHz s$^{-1}$ by Menietti et al. (2000). Striations in SKR seem to last longer than in AKR. Most AKR

striations are shorter than 3 s (Menietti et al., 2000), whereas a first estimation for the durations of striations in SKR shows that they typically last around 5–10 s as can be seen in the right panel of Figure 4. This might be due to the more extended magnetosphere of Saturn in comparison to Earth, and thus longer propagation distances for EMIC-waves or ion holes. A more detailed comparison of AKR with SKR striations and other fine structures is planned for a future paper. It is currently not known why striations tend to become interrupted lines at medium frequencies and appear as rain as shown in Sect. 4. Mutel

et al. (2006) suggested that upward traveling ion holes are the cause for AKR striations.





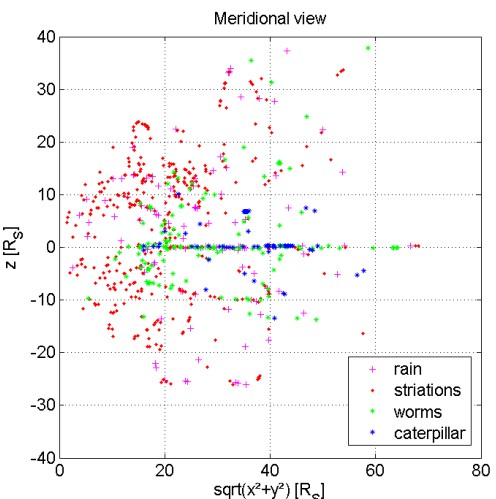
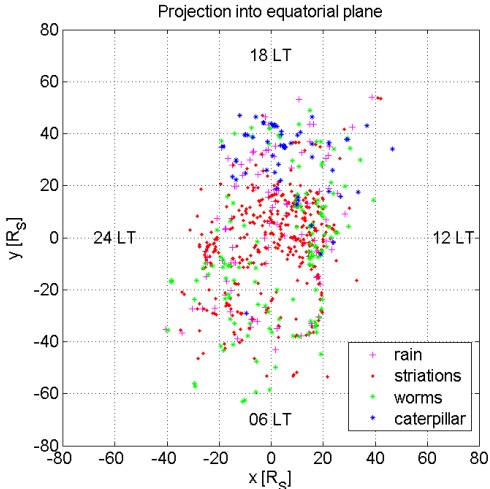

**Figure 8.** Meridional and equatorial distribution of SKR from the SPEC class (rain, striations, worms, caterpillar) below 80 kHz on the left and right side, respectively.

The special structures of worms and caterpillar have been newly introduced in this paper and were never described before. A worm could be interpreted as a source oscillating in altitude. Caterpillars are not only a structure with a relatively smooth distribution of intensity, but due to their long duration they should actually rather be called a coarse structure. Worms were almost exclusively found in 80 kHz spectra and very rarely at higher frequencies, and most caterpillars were found below

40 kHz, and typically had a center frequency around 10 kHz. They somewhat resemble Saturn narrowband emissions as they also have a constant frequency over several hours. However, the bandwidth of caterpillars is usually around 10 kHz, whereas narrowband emissions only go over 2–3 kHz (Wang et al., 2010), and this difference can be seen when comparing the caterpillar on the right side of Figure 1 with the narrowband emissions on the left side of the same figure. Furthermore, the polarization properties of caterpillars are rather similar to SKR than to narrowband emissions, and therefore we consider them as a part of

SKR. This should also be investigated in more detail in a future paper. The constant frequency of a caterpillar would imply a relatively constant source at a constant altitude, and the long duration suggests that this source extends over several hours in local time. This can be said because the rotation of the planet would rotate a single source magnetic field line and the corresponding beamed emission out of view of the spacecraft within several hours.

We also investigated the occurrence of various fine structures as a function of the position of the spacecraft. No dependence

on the spacecraft position was found for dots (DOTS), areal features (AREA), and linear structures (HORZ, VERT, NEGS, POSS). However, structure from the class of special features (SPEC) do show some dependence, and the meridional and equatorial distributions of rain, striations, worms, and caterpillars below 80 kHz are shown in Figure 8. In the meridional view on the left side it can be seen that rain and striations can be observed close and far from the planet, but tend to be more common at medium and high latitudes compared to low latitudes. This is just the opposite for worms and caterpillars which rather occur



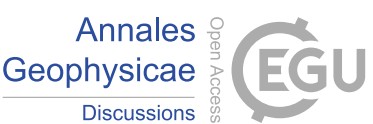

at low to medium latitudes and in the equatorial plane. Furthermore, no caterpillars were observed at distances closer than
$\sim 15\,\mathrm{R_S}$, and almost no worms were observed closer than $\sim 10\,\mathrm{R_S}$. These distances are far beyond the equatorial shadow zone
of SKR within $3.7\,\mathrm{R_S}$ (Lamy et al., 2008), and it is unclear why worms and especially caterpillars were not observed close
to the planet. The equatorial distribution on the right side of Figure 8 shows a slight preference of rain and striations to be
observed from the afternoon and dusk side, but principally they can occur at any local time. Worms do not seem to show any
preference for local time, but it is obvious that caterpillars are almost exclusively observed from local noon to dusk with a
clear preference for the range of 16–20 LT around dusk. A potential reason for this could be that caterpillars rather need stable
magnetospheric conditions at their source field lines to ensure their frequency stability, and such a stability is not given at the
morning side where Kelvin-Helmholtz instabilities can occur at the magnetopause and influence the acceleration of electrons
into the auroral region (Masters et al., 2010).

Finally, we want to add which special structures found in terrestrial or Jovian radio emissions were not found in SKR spectra.
The banded emissions in AKR separated by the ion cyclotron frequency as shown by Grabbe (1982) have no counterpart in
SKR. Similarly, the zebra patterns found in Jovian broad-band kilometric radiation by Kurth et al. (2001) were not seen in any
of the thousands of SKR fine structure spectra which we classified.

## 6   Conclusions and outlook

This paper made a first attempt to classify fine structures of auroral radio emissions according to their geometrical shape in
the time–frequency plane. Our classification scheme was introduced for Saturn kilometric radiation (SKR) observed by the
Cassini RPWS Wideband Receiver (WBR), but it could be adapted also for other planets like Earth or Jupiter. The classes of
dots, areal features, and linear features should be rather universal, but different special structures could be introduced at any
other planet with auroral radio emissions. Here we newly introduced the special structures of worms (which might be SKR
sources oscillating in altitude) and caterpillars. The latter have a constant frequency below $\sim 40\,\mathrm{kHz}$ for several hours, a typical
bandwidth of $\sim 10\,\mathrm{kHz}$, and they tend to be observed beyond distances of $\sim 15\,\mathrm{R_S}$ around local dusk. The fine structures of
rain and striations have also been observed for AKR, and here we defined the structure of rain as interrupted striations which
rather tend to occur at medium frequencies around 325 kHz compared to low frequencies below 80 kHz. Especially striations
and caterpillars would require a more detailed investigation in future papers. Furthermore, linear features and especially lines
with positive slope were found to be more common at 325 kHz compared to below 80 kHz, which might give some clues about
the generation of fine structures for which a large number of theories exist. We also observed rare instances of striations with
a positive slope, and rare absorption signatures within areal features.

*Data availability.* This research is based on the Cassini Radio and Plasma Wave full resolution wideband data, which can be found in the
NASA Planetary Data System (Kurth et al., 2018) under https://doi.org/10.17189/1519615. Reprocessed WBR spectra in new design can be
found at http://babeta.ufa.cas.cz/cas_wbr_ql/.



*Author contributions.* G.F. wrote the paper, introduced the classification scheme, and classified the SKR fine structures of the Cassini Wide-band Receiver (WBR). U.T. and D.P. processed the WBR data, prepared newly designed dynamic spectra, and contributed to their classification. All authors discussed the results and commented on the manuscript.

*Competing interests.* The authors declare that they have no conflict of interest and no competing interests are present.

*Acknowledgements.* This research was funded in whole by the Austrian Science Fund (FWF) [I 4559-N] and by the Czech Science Foundation GAČR [20-06802L], and G.F., U.T, and D.P. acknowledge support from the FWF-GAČR international project "Analysis of fine structures in auroral radio emissions". For the purpose of open access, the author has applied a CC BY public copyright license to any Author Accepted Manuscript version arising from this submission.



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
