# Peer review of "Classification of spectral fine structures of Saturn kilometric radiation"

_Annales Geophysicae, 2022_

## Author Response (AR1)

Dear Editor and Reviewers,

I send you the revised version of our manuscript on „Classification of spectral fine structures of Saturn kilometric radiation". In this document you will find again your constructive reviewing comments (in black italic font) and our replies (in blue color).

Best regards,

Georg Fischer (on behalf of all co-authors)

*Reviewer 1:*

*Classification of spectral fine structures of Saturn kilometric radiation by Fischer et al. is an interesting paper that characterizes the spectral fine structures of Saturn kilometric radiation (SKR) observed by the Wideband Receiver of the Cassini's Radio and Plasma Wave Science instrument. The authors introduce seven different classes of SKR fine structures 1)dots (one class for 0-dimensional objects), 2-5) lines (1-dimensional objects being horizontal, vertical, or with negative or positive slope), 6)areal features (2-dimensional objects), 7)special structures named according to their appearance in time–frequency spectra (rain, striations, worms, and caterpillar). The authors define the various characteristic (for example the frequency width and slope, and emission duration) for each classification. The authors then examine the statistics of the occurrence of the various classes and sub-classes and their characteristics and discuss the implications in view of the many theories about spectral fine structures of auroral radio emissions.*

*This paper is well written, the abstract and presentation is clear and understandable, the figures are of high quality and clearly show the data and results, and the references are comprehensive.*

*The paper is primarily an observational work (characterizing the spectral structure of the SKR) with some discussion on how the observed structure fits into the various theories of SKR generation and propagation. I rate the scientific contribution as "fairly important" and believe the paper should be published.*

Dear reviewer,

thank you very much for your review, and we see that you are mainly okay with it. You are right that our work is mainly an observational work characterizing the spectral structures. We only would like to add that (surprisingly) it is the first time that such a kind of classification has been made.

No request for changes of the manuscript were made in this first review.

*Reviewer 2:*

*The paper studies fine structure of Saturn kilometric radiation (SKR) using data obtained by the Cassini Wideband Receiver (WBR), a part of the Radio and Plasma Wave Science (RPWS) instrument. These features are believed to result from physical processes associated with the generation of SKR, presumably by the cyclotron maser instability. Seven different classes of fine structure are identified in detail, ranging from dots, lines with different slope, to areal stuctures. A 7th class identifies special structures based on unique morphologies on f-t spectrograms. Carefully*

*conducted statistical studies of the occurrence of these different classifications yield a number of intriguing results, which the authors discuss in great detail. For instance, lines with a positive slope are much more common at medium frequencies (near 325 kHz) than at low frequencies (below 80 kHz), and vertical lines are almost absent at low frequencies. "Striations" (group of narrowbandedlines with predominantly negative slopes) are quite common below 80 kHz, but less common near 325 kHz. At medium frequencies striations appear a "rain" or interrupted striations. Sub-classifications of "worms" and "caterpillars" are formally discussed for the first time. The authors present a long discussion of the results of the survey in light of the CMI, and possible explanations (or lack thereof) for a number of the classifications.*

*We find the work to be well-written with an extensive Introduction briefly discussing the cyclotron maser instability and observations of AKR and SKR along with a review of finestructure observed at Earth and Saturn to date. While there are no scientific conclusions explicitly given in this work, the analysis is essentially a necessary first step in the process of understanding the physical explanation for the "zoo" of finestructure observed resulting from the CMI.*

*Some highlights:*

*>Excellent description of the RPWS WBR along with modes of operation.*

*>Detailed description of the 7 types of classifications. Occurrence probabilities are described for each category in detail, including how some complications were handled, such as dots associated with rain or striations.*

*>Detailed Discussion of the emissions where a number of questions arise, such as identifying the not always definitive modes of the emission.*

Dear Reviewer,

thank you for your careful comments and positive evaluation of our work. Below you can find answers to your questions.

Best regards,

Georg Fischer (for all co-authors)

*Here we list some enumerated comments and a few questions.*

*L 79--Please define "areal" in this context.*

We removed the words with "areal" within the brackets, because it should not be necessary to give examples. So the sentence now simply reads as: "Section 5 will discuss some aspects of various classes with respect to physical characteristics or generation hypotheses."

*L 157--Replace "own" with "unique"?*

Done as requested.

*L 180--Have you checked for instrumental signatures for the VERT signals, perhaps due to another instrument starting or changing modes, etc.? [I see you do discuss this later near Lines 397-399]*

This is a difficult issue, because we are not aware that there is a comprehensive list of all Cassini instruments with all mode changes and possible electromagnetic interferences. That is why we choose a cautious approach in Lines 397-399. However, we are now more confident that most VERT signals are real, because of the following reasons: Not all VERT signals go over the whole bandwidth from 310-340 kHz, and many are shorter in bandwidth (but at least go over 10 kHz according to our selection criteria). So the right panel of Figure 2 might be not such a good example, and we replaced it by one showing a VERT signal with a bandwidth of about 20 kHz. This signal should not be caused by a current surge in the spacecraft electronics as such signals are very broadband. Additionally we checked many VERT signals also in low temporal resolution HFR plots, and we found no interference signals extending over a large bandwidth. This also suggests that many of them are real signals. The fact that almost no vertical signals are found below 80 kHz strengthens the hypothesis of Gurnett et al. (1979) that the frequency drifts are due to source movements (we added this sentence at the end of the paragraph). Such movements over a bandwidth of 30 kHz are physically possible at 325 kHz, but not at 80 kHz since the source speeds would be too high. We changed at line 397: "..., and therefore it cannot be excluded that some vertical lines at 325 kHz are caused by spacecraft interferences. However, vertical lines caused by current surges should have a large bandwidth, and many vertical signals around 325 kHz do not extend over the whole bandwidth of 30 kHz, which suggests that at least they are real natural signals."

L 177-Could this feature be Z-mode or O-mode?

First of all, we would like to mention that we are not aware of any publication identifying Z-mode in SKR. Furthermore, the WBR operates with just one antenna, so that it is impossible to determine the polarization (which would give us a hint on the mode) for features with a small extension in the time-frequency plane. Nevertheless, we tried to look at the polarization for the features of Figure 2 (left panel) by looking at the RPWS-HFR data, which has a temporal resolution of only 16 seconds. We carefully looked at the polarization spectrum in the right places of the features (HORZw, NEGSn) and found that they should have a RH polarization. Cassini is actually very close to the equatorial plane on DOY 250, 2006 around 10 SCET (1 deg N), so that SKR from both hemispheres is recorded, and the polarization spectrum shows a mix of RH and LH SKR. Hence we cannot tell if the RH SKR is R-X-mode SKR from the North or L-O-mode SKR from the South. Both options are possible. We think that we can exclude the Z-mode since the local cyclotron frequency fc and the local plasma frequency fp are way too low for the Z-mode to propagate at 40 kHz at a distance of 23.6 Saturn radii, where the spectrum was recorded by Cassini.

*Figure 4, right side: How about the 3 or 4 narrow features that have the*

*shape of a shallow upward parabola? Are these POSSn?*

No, these upward parabolas are spacecraft interferences. They can also be seen on the left panel of Figure 1 (marked as interferences), the left panel of Figure 2, the right panel of Figure 3 (also marked as interferences), and on both panels of Figure 5. In all cases they last exactly for 16 seconds, so they cannot be of natural origin. We do not exactly know what causes them, but one possibility would be the HFR frequency sweep: This often lasts 16 seconds and for all examples in the figures the start of the upward parabola happens at the same time as the start of the sweep. We modified the sentence describing Figure 1, where those artificial features first appear: "The narrowbanded drifting tones occurring every 15-20 kHz are marked as interferences, and their regular appearance and constant duration of 16 seconds suggests that they are articifical signals from the spacecraft." (Line 149)

*L247-248: Do you mean "...but we count only the number of spectra WITH classified linear, areal or special structures"?*

No, here we are talking about which spectra we classify and count as UNCL (unclear). We re-wrote the sentence in the following way to make this more clear: "However, for the count of the number of unclassified spectra (UNCL) we only take those without any classified linear, areal or special structures."

*L352--"...is one of THE areal features..."*

Corrected as suggested.

*L374-377-These "absorption features" might perhaps also be refractive attenuation signatures as discussed by Gurnett et al., 1998 (GRL, doi:10.1029/98GL01400) for Jovian hectometric radiation. The refraction is due to grazing incidence of HOM near the edge of the Io torus. Gurnett et al., referred to these features as "attenuation bands" (see their Figure 3).*

This might be possible, although it should be said that the HOM absorption features at Jupiter occur in the MHz range and have a much larger bandwidth (100 kHz) and duration (hours) compared to what we see in SKR at Saturn around 40 kHz with bandwidths of a few kHz and durations of a few minutes. Nevertheless, we added the following sentence at the end of this paragraph: "Another option would be that absorption features perhaps might be refractive attenuation signatures due to grazing incidence of SKR near the edge of the Enceladus plasma torus (Persoon et al., 2009), somewhat similar to attenuation lanes in hectometric radiation at Jupiter (Gurnett et al., 1998)."

*L440-443: The oscillating worms may be the result of the source region plasma characteristics oscillating. Density gradients near or within the edge of the Enceladus torus may slowly change with longitude as an extended source region rotates into or out of view. This might also be consistent with your study of the position of the features in the Saturn magnetosphere.*

We are sorry, but it is difficult to follow your argument here. SKR is no plasma wave emission, so that a change in the source region plasma density should not have an effect on the frequency of the emission. The plasma density in the source region might influence the growth rate of the CMI

mechanism creating the SKR, but not its frequency. The SKR source should also not be at the edge of the Enceladus plasma torus, but at much higher latitude (roughly around 70-75 deg. North or South) as shown by studies of Lamy et al. (2009, 2013) who found SKR on magnetic field lines co-located with auroral UV features. For examples, the worms of Figure 3 with a frequency of 70 kHz might be around a latitude of 75 deg. S at a distance of 2.5 Saturn radii from the center of the planet, which is nowhere near the edge of the plasma torus. So no change was made to the manuscript regarding this issue.

Finally, we would like to mention that we corrected an error related to the meridional and equatorial distribution of the special classes (rain, striations, worms, caterpillar) in Figure 8. We had just displayed one classification per spectrum, and had not taken into account that one spectrum can have several classifications. This is now corrected, and the meridional view looks quite similar, only more data points are present. However, in the equatorial view the preference of caterpillars for the 16-20 LT region disappeared, and all features are more or less uniformly distributed in local time. There is only the region from 20-24 LT in which all special features seem to be somewhat less common. This might be related to the preference and large intensity of SKR sources in the local morning from where they can be beamed to most local times, but not to the pre-midnight section. The figure, its caption, the text in the discussion (around line 460), abstract (caterpillars now down to 10 R_S) and conclusion were modified accordingly. Please also note that the right panel of Figure 2 was modified to show a vertical signal (VERT) that extends over 15-20 kHz in bandwidth. The main text (around line 180) was also modified accordingly.